# A Regulatory Network for miR156-SPL Module in *Arabidopsis thaliana*

**DOI:** 10.3390/ijms20246166

**Published:** 2019-12-06

**Authors:** Chenfei Zheng, Meixia Ye, Mengmeng Sang, Rongling Wu

**Affiliations:** 1Center for Computational Biology, College of Biological Sciences and Technology, Beijing Forestry University, Beijing 100083, China; zhengchenfei@163.com (C.Z.); yemeixia123@bjfu.edu.cn (M.Y.); sangmm12345@163.com (M.S.); 2Center for Statistical Genetics, Pennsylvania State University, Hershey, PA 17033, USA

**Keywords:** phase change, miR156, next-generation sequencing, flowering plant, gene regulatory network

## Abstract

Vegetative phase changes in plants describes the transition between juvenile and adult phases of vegetative growth before flowering. It is one of the most fundamental mechanisms for plants to sense developmental signals, presenting a complex process involving many still-unknown determinants. Several studies in annual and perennial plants have identified the conservative roles of miR156 and its targets, *SBP/SPL* genes, in guiding the switch of plant growth from juvenile to adult phases. Here, we review recent progress in understanding the regulation of miR156 expression and how miR156-SPLs mediated plant age affect other processes in *Arabidopsis*. Powerful high-throughput sequencing techniques have provided rich data to systematically study the regulatory mechanisms of miR156 regulation network. From this data, we draw an expanded miR156-regulated network that links plant developmental transition and other fundamental biological processes, gaining novel and broad insight into the molecular mechanisms of plant-age-related processes in *Arabidopsis*.

## 1. Introduction

The transition of vegetative growth from the juvenile to the adult stage is essential for sexual reproduction [1]. Many plants change vegetative traits with the development of their ability to flower, as differentiation pattern distinct between adult and juvenile organs [2]. Understanding the mechanisms that regulate adult vegetative differentiation to develop floral competence from the juvenile phase remains a fundamental challenge in plant biology [3].

Recently, molecular techniques have been widely used in studying the genetic mechanism of vegetative phase transition, from which microRNA156 (miR156) and its target *SBP/SPL* genes were detected to be major regulators for plant development [3,4]. It has been recognized of the extremely conserved role of miR156 plays throughout the angiosperms, as evidenced from many species [5,6,7,8].

Despite the availability of several seminal reviews illustrating the basic function of miR156 in developmental transitions [2,4], an overall picture of its controlling mechanisms in a regulatory network remains to be elucidated. Here, we provide an overview of how miR156 affects phase change through a network mainly in *Arabidopsis*. We pinpoint a few key genes and transcription factors that regulate and/or are regulated by the expression of miR156 during phase change. The recent years have witnessed a great advance in using powerful deep sequencing analysis to identify the miR156-SPL module involved in many other processes [9,10,11,12,13,14,15,16,17,18]. Based on the available knowledge from both experimental and sequencing data, we illustrated a miR156-mediated regulation network that helps us understand the molecular mechanisms behind the plant-age-mediated phenomenon.

## 2. Molecular Regulation of Phase Change

### 2.1. miR156 As a Key Regulator

Since miR156 was first discovered to modulate vegetative phase transition in *Arabidopsis* and maize, its regulatory role has been widely observed in other plants [5,7]. This microRNA accumulates at the highest level in the seedling stage, followed by a gradual decline with plant development. Plants that contain rich miR156 have a prolonged juvenile stage, which is characterized by increased branching, accelerated leaves emerging and retarded flowering [6,8,19]. In *Arabidopsis*, this so-called “heteroblastic” development is marked by increased margin and complexity of leaves and decreased cell size [4]. These changes that occur within the same plant are attributed to sequential alterations in miR156 and their squamosa promoter binding protein-like (SBP/SPL) targets [4].

### 2.2. Regulation of miR156 by Endogenous Cues

#### 2.2.1. Fine-Tuning Regulation of miR156 by GCT/CCT

It has been recognized that the continuous decline of miR156 with age promotes the emergence of the adult stage. How age regulates the temporal decrease of miR156 expression has been documented in different studies [20,21]. Gillmor et al. (2014) [22] found that two additional upstream regulators of miR156, *GCT* (*Arabidopsis MED12*) and *CCT* (*MED13*), presenting two elements of the mediator CDK8 module, act as global regulators for the developmental transition from germination to flowering (Figure 1a). Northern blotting analyses indicate that miR156 levels are double in *gct* and *cct* than in wild type (wt), whereas the expression level of *SPL3* and *SPL9* are repressed in these mutants. A single copy of 35S::MIM156 transgenic plant produces abaxial trichomes in leaf 1, but not in leaf 15 and 16 in both *gct* and *cct* background. Although completely epistatic to *gct* and *cct* for producing trichome, transgenic plants with single 35S::MIM156 do not show complete suppression of leaf serration and length/width ratio. This result suggests that *gct* and *cct* have functions during leaf development through miR156-dependent and independent functions [3,22].

#### 2.2.2. Sugars Repress Expression of miR156 Genes

It is hypothesized that sugar was required as fundamental energy for larger size and more complex structures of adult leaves. Yu et al. [20] and Yang et al. [21] independently proved the vital role of sugars in promoting the transition of plant growth by repressing the transcription of *miR156* gene into messenger RNA (Figure 1b). In plants whose leaves are removed and, therefore, sugar abundance is reduced, miR156 is found to be upregulated, in a cope with the delayed juvenile-to-adult transition. Glucose not only suppresses miR156 transcription but also breaks down their primary mRNA transcripts through Hexokinase-1 (*HXK1*). However, plants lacking *HXK1* show slight progression into the mature form, indicating that *HXK1* is not solely responsible for vegetative phase change. This role of sugars in miR156 modulation is evolutionarily conserved.

#### 2.2.3. Epigenetic Regulation of miR156 Genes

Once perennial plants transition into the adult, they do not rejuvenate without exogenous influences, meaning the transition is unidirectional. How plants preserve the stability of each phase has long been of interest to plant biologists. Recent results by Xu et al. (2015) [23] indicate the important roles of changes in chromatin structure in the vegetative phase transition. They found an accumulation of histone H3 lysine 27 trimethylation (H3K27me3) is cooperating with a decline in the expression level of *MIR156A/MIR156C* and with an increased amount of PRC2, which is regulated by the E (z) homologs SWINGER and CURLYLEAY. Furthermore, the rate of H3K27 acetylation is regulated by PKL, which acts as a member of the CHD protein subfamily II. According to these results, Xu et al. (2015) [23] proposed a hypothesis to illustrate the molecular regulation of vegetative phase change: at the juvenile stage, H3K27ac of *MIR156A/MIR156C* is higher enough to prohibit PRC2 binding. Then, PKL bind to the promoter of miR156 genes and reduce the H3K27ac level, cooperating with a histone deacetylase (X). As plants transition to the adult phase, the affinity of PRC2 for *MIR156A/MIR156C* raises either by no longer inhibition caused by lower H3K27ac levels or by the assistance of a temporally regulation factor (Y). Then H3K27 is methylated by PRC2 of *MIR156A/MIR156C* and repress transcription of these genes, resulting in a decline of miR156 level during the vegetative phase transition (Figure 1c).

Very recently, more pieces of evidences of epigenetic regulation of miR156 genes expression have been documented [24,25]. These transcriptional regulations of chromatin mainly based on both covalent and noncovalent modification of histones. SWINGER (SWN), one of the most important methyltransferases, participated in the trimethylation of H3K27me3, which is an important covalent modification of histones in plants. A recent work indicates that SWN promote vegetative phase change by repression expression of miR156 genes through the trimethylation of H3K27me3 [23]. BRAHMA (BRM), a gene-encoded type of SWI/SNF-type chromatin complex subgroup protein, is the most widely studied DNA-dependent ATPase, mediating noncovalent modification of histones. The role of BRM in vegetative phase change remained unknown until a recent work [24]. A single *brm* mutant promotes vegetative phase change, indicating that BRM functions as a repressor for vegetative phase change. Overexpression of miR156 genes could rescue these precocious phenotypes in *brm* mutant background. BRMs directly bind in the promoter region of miR156A, downregulating its expression. A significant increase of nucleosome occupancy signal at a proximal region of *MIR156A* gene is detected in a single *brm* mutant, indicating BRM active MIR156A transcription by decreasing nucleosomes occupancy rather than by shifting nucleosome position. In addition, single mutant *swn*, but not *clf*, could partially rescue the precocious phase change phenotype in the *brm* background. As an important methyltransferase, SWN contributes to the increasing level of H3K27me3 at the promoter region of *MIR156A* locus, maintaining a relatively stable expression level of miR156 when plants age. Taken together, a SWN-BRM antagonistic interaction model is proposed for the expression pattern of miR156A during vegetative phase change. When plants develop, both SWN and BRM act normally but antagonistic to maintain the expression level of miR156. SWN function as methyltransferase, leading to the elevated level of H3K23me3 to downregulate *MIR156A* expression, but BRM increases the occupancy of nucleosomes at the promoter region of *MIR156A* locus, which represses miR156 expression. BRM binds to the promoter of MIR156A locus to reduce the nucleosome occupancy, meanwhile, BRM antagonizes SWN to reduce the level of H3K27me3 at the *MIR156A* locus. This interaction activates *MIR156A* expression when the plant is juvenile. Then plants age, and the *MIR156A* expression is repressed due to the elevating level of H3K27me3 caused by SWN overtakes the function of BRM, although BRM still functions normally (Figure 1d). However, the transcription factors that directly mediate the expression of *MIR156A* must be identified in the future.

Another ATP-dependent chromatin remodeling complex, SWR1 could replace H2A with H2A.Z variant. As one of the three important components of SWR1 complex, loss of function mutant *arp6* contributes to the balance of miRNAs and their target mRNA during some development process, such as flowering time and leaf morphological variants [26]. Furthermore, SWR1 also contribute to the initially high expression of miR156 genes during vegetative phase change by exchange histone variant from H2A to H2A.Z [25]. Loss-function mutants of the two components (ARP6 and SEF) of SWR1 complex accelerate vegetative phase change by repressing miR156 gene expression. A similar phenotype is also detected in the mutants of H2A.Z encoded genes. Significant reduced level of miR156 expression is detected at the early stage of juvenile phase in both *arp* and *sef* mutants. Additionally, the abundance of H2A.Z at miR156 locus decreases significantly in *arp* mutant seedlings. These results suggest that ARP6 and SEF promote the juvenile phenotype through their effect on H2A.Z. Then they observed the elevated level of H3K4me3 at both *MIR156A* and *MIR156C* promoter region, and the level ofH3K27me3 only reduced at *MIR156A* region in *arp6* mutant. However, no significant signals of MNase (micrococcal nuclease) sensitivity are detected at *MIR156A* or *MIR156C*, except at a site of the +1 nucleosome in *MIR156A*. Taken together, H2A.Z promotes miR156 gene expression mainly by its effect on H3K4me3. In addition, a H3K4 methyltransferase, ATXR7 participate in the deposition of H3K4me3 in regulating the expression of miR156. Based on these results, they proposed that SWR1 complex contribute to the initial high expression of miR156 by exchanging histone variant of H2A.Z. Then H2A.Z recruits H3K4me3 deposition at the promoter region of miR156 locus, leading to the high expression of miR156 genes at the initial stage of the vegetative phase change (Figure 1e). However, the variation of H2A.Z or SWR1 complex does not account for the successive decline of the abundance of miR156 during shoot development.

## 3. The miR156-SPL Act as a Regulatory Hub

### 3.1. miR156-SPL Regulates Developmental Process

#### 3.1.1. miR156-SPL Regulates Flowering Time and Reproductive Organ Development

##### miR156-SPL Regulates Flowering Time

The so-called “age pathway” illustrated a specific flowering regulation process mediated by miR156-SPL module, which was first detected by Detlef Weigel and his colleagues [27]. Under both long- and short-day conditions, flowering time is delayed by overexpression of miR156. Another conserved miRNA, miR172 acts as a flowering promoter by repressing its targets, AP2-likes genes which inhibit the expression of *FT* [1]. By contrast to the role of miR156, overexpressing of miR172 induce flowering much earlier. As demonstrated by chromatin immunoprecipitation results, miR172 is directly activated by SPL9, promoting juvenile–adult phase transition in *spl* transgenic plants [7,28]. This interactive between miR156-SPL and miR172-AP2 modules act as central regulators in juvenile–adult–reproductive phase transition (Figure 2a), which is conserved throughout the plant kingdom [6,8,29].

##### miR156 Regulate the Distribution of Trichomes on the Inflorescence Stem

Trichomes act as a fundamental barrier protects plants from exogenous damage such as herbivores, UV irradiation [30]. During the vegetative phase in *Arabidopsis*, trichomes only distribute on the adaxial (upper) surface of the leaf blade when plant is at juvenile phase (hereafter trichomes on the leaf surface is termed as the leaf hairs). Then initiation of leaf hairs on the abaxial (lower) surface of leaf illustrates plant transiting into adult phase. After entering the reproduction stage, the numbers of trichomes on the inflorescence stem reduce progressively. These facts demonstrate that the distribution of trichomes is regulated temporally and spatially. The temporal regulation of trichomes distribution is mediated by directly activate of two MYB transcription factor genes, i.e., *TCL1* and *TRY*, which repress trichome development. SPL9 activates the *TCL1*/*TRY* gene expression by directly binding to the promoter region. When *Arabidopsis* ages and become ready for flowering, reduction of miR156, releasing more SPL9, bind to the *TCL*/*TRY* promoter to activate their expression, leading to the decreased number of trichome on inflorescence stem (Figure 2b). This process is independent of the GLABROUS1 (GL1) and GIS-mediated phytohormone pathway, although they also affect the trichome formation in *Arabidopsis*.

##### miR156 Secure Male Fertility

Specialized organs, including stamen and carpels, are essential for sexual reproduction after plant entering the reproductive phase marked by the formation of flower. In *Arabidopsis*, the development of stamen can be divided into 14 different stages following stamen identity and determining of stamen identity is mediated by SPOROCYTELESS/NOZZLE (SPL/NZZ). In addition to SPL/NZZ, a member of SPL family, SPL8, which is not the direct target of miR156, also participates in anther development. Overexpression of miR156 leads to nearly full sterility, whereas miR156-targeted the SPLs partially to “rescue” the fertility problem in the *spl-8* mutant. This result illustrates at least one miR156-targeted SPL function redundantly with SPL8 to maintain fertility. Additionally, miR156-targeted SPLs interact with SPL8 mainly affect sporogenous cell proliferation in early anther development. Interestingly, the expression of miR156-targeted SPLs gene is regulated temporally and spatially at different stages of anther development. For example, expression of *SPL11* is detectable in the anthers at stage 2 and 3, while the strongest expression of *SPL11* is detected in the pollen mother and tapetal cells at stage 5 of anther development. Contrasting to the expression pattern of *SPL11*, the strongest expression of *SPL13* is detected in tapetum at stage 5 anther. Furthermore, SPL/NZZ represses the expression of miR156-targeted SPLs through promoting miR156 expression in inflorescences whereas miR156-SPLs seem to have no influence on the *SPL*/*NZZ* expression (Figure 2c). According to these results, early anther development in *Arabidopsis* is mediated by both miR156-targeted SPLs and non-targeted SPL8, interacting with SPL/NZZ or independently regulating cell division and differentiation to form proper cells, which is essential for anther development [31]. Obviously, more detailed knowledge of this model is still required, especially of the direct target of miR156-SPL module.

#### 3.1.2. miR156 Mediate Heteroblastic Change of Leaf Morphology

The gradual change of several morphological traits, such as leaf size and leaf shape along the shoot of a plant is termed as heteroblasty. In *Arabidopsis*, this heteroblastic development involved in the ration of leaf length to width, leaf serration and complex and the emergence of trichomes (leaf hairs) on the abaxial surface of leaf blades [32,33,34]. The transition from juvenile to adult phase is marked by these heteroblastic traits. The mechanistic connection between vegetative phase change and heteroblastic development of leaf traits has been recently documented, which is mediated by the miR156-SPL module.

##### miR156-miR319-miR164 Coregulate Leaf Morphological Change

Rubio-Somoza et al. (2014) [32] reported that the developmental and morphological changes of leaf are mechanistically linked by microRNAs and their targets, which explains the increased serration and complexity of consecutive leaves, a phenomenon pervading the plant kingdom. The expression pattern of cup-shaped cotyledon (CUC) transcription factors for organ initiation and demarcation determine leaf serrations in *Arabidopsis thaliana* and leaflets in *Cardamine hirsute*, a relative of *A. thaliana*. Overexpression of miR164 decreases the depth of serration through inhibiting CUC2 activity. In addition, miR319-regulated Teosinte branched 1/cycloidea/PCF (TCP) also plays a role in leaf serration formation. Rubio-Somoza et al. (2014) [32] demonstrating that TCP4 prevents dimerization of CUC2–CUC3 by heterodimerization with both CUCs, reducing leaf serration and complexity in younger leaves. But as leaves age, SPL9 competitively dimerizes with TCP4 and, therefore, activates the heterodimerization of CUC2 and CUC3, promoting leaf serration and complexity (Figure 2d). This process is under the control of SPL, the target of miR156, which is the master determinant of the change of leaf shape.

##### miR156 Mediate the Leaf Hairs Initiation

When *Arabidopsis* enter the adult phase, epidermal leaf hairs emerge at the lower surface of leaf blades. Abaxial hairs are only observed after the sixth leaf of wild-type *Arabidopsis* under long-day conditions, suggesting that both plant age and leaf polarity attributed to this trait. However, the mechanical connection between plant age and leaf polarity was still unknown until recently in two independent works [33,34]. A dominant mutant of *GLABRA1* (*GL1*) gene, *gl1-D* accelerates the initiation of leaf hairs caused by a 3′ noncoding region substitution (G to A) in the GL1 gene [33]. TOE1, an AP2-like transcriptional factor is identified binding to the 3′ region of *GL1* to repress its expression by both works. The interaction between a leaf polarity regulator (KAN1) and TOE1 is verified by yeast assays in vivo and by bimolecular luminescence complementation (BiLC) assays [34]. Furthermore, KAN1 targets *GL1* by binding to the 3′ nocoding region of *GL1* near its enhancer. Taken together, KAN1 interacts with TOE1, then they both bind to *GL1* 3′ noncoding region to repress *GL1* expression. The repression of *GL1* by KAN1 and TOE1 is dependent on the chromatin looping mediated by enhancer of *GL1* gene. In addition, KAN1-TOE1 represses *GL1* expression with the help of TOPLESS (TPL) mediated deacetylation of histone in the juvenile phase. Based on their result, the epidermal leaf hairs initiation on the lower surface is repressed in juvenile phase. The KAN1-TOE1 complex binds to the 3′ noncoding region, facilitating the formation of a chromatin loop at the *GL1* locus. Then TPL was recruited by TOE1, reducing the levels of acetylation of histone H3K9Ac and H3K14Ac, leading to the repression of *GL1*. When plants age, the accumulation of miR172 elevates and represses its target, TOE1. TPL no longer exists at the *GL1* locus, and the acetylation levels of both H3K9Ac and H3K14Ac elevated, resulting in the activation of *GL1* gene to eventually form leaf hairs at the abaxial surface of the leaf blade (Figure 2e).

#### 3.1.3. miR156 Regulate Root Development

Root system architecture is a fundamental organ for plants, which consist of lateral root and root hair in dicotyledonous plants like *Arabidopsis*. Several miRNAs (miR160, miR164, miR165/miR166) involve in regulating lateral root development through the auxin signal pathway [35,36,37], nutritional homeostasis and response to environmental stress. However, the role of miR156 during root development remained unknown until recent works. Overexpression of miR156 gene produced more lateral root and vice versa. Three types of miR156-targeted SPL genes (*SPL3*, *SPL9*, and *SPL10*) also involved in the lateral root development. Among these SPL modules, SPL10 play the dominant role, with SPL10 function high activity in the all kind of tissue in primary roots as well as the entire stage of lateral root development. Furthermore, nearly no lateral root arises even after 15 days in the *spl10* transgenic line. The miR156-SPL module regulates the abundance of lateral root mainly through its effect on the primordia progression rather than that on the meristem size of lateral roots (Figure 2f). Moreover, this miR156-SPL module response to the auxin signal transduction pathway. Treatment of IAA could induce the expression of miR156 gene (*miR156B* and *miR156D*) and repress the gene expression of miR156-targeted SPLs (*SPL9* and *SPL10*). These results demonstrate a connection between vegetative phase change mediated by miR156-SPL module and auxin signal transduction in regulating lateral root development [38]. Another conserved miRNA, miR164, which targets NAC1 to have a role in lateral root development by transducing auxin signal. Does the miR164-NAC1 module affect miR156-SPLs during root development? What are the downstream targets of SPL10 during lateral root development? Does miR156-SPLs have a role in the emergence of root hair? These questions need to be addressed in the future.

### 3.2. miR156-SPLs Regulates Response to Environment Stress

Unlike animals, as sessile organs, plants have to detect and respond to environmental stress to ensure their survival [39,40]. Once plants are confronted with adversity, the development process is delayed. When the environment recovered, the development continued. This relationship between development and stress response is thought to be the resulting from the rebalancing of energy assignment. The molecular mechanism underlies this balance has been verified to be mediated by the miR156-SPL module [39,40].

#### 3.2.1. Inducing of miR156 in Response to Abiotic Stress

Younger plants seem to be more tolerance to environment abiotic stress. When plants face up to adversity, flowering will be delayed, ensuring their survivor. Expression of miR156 genes (mainly *MIR156A* and *MIR156C*) will be induced under various stress conditions (salt and drought) prolonging the juvenile phase to help plants withstand the unfavorable environments. When the conditions recovered again, the accumulation of miR156 decrease, preparing for flowering. Accumulation of anthocyanin also increases during the stress response. This results suggesting miR156-SPLs coordinate development and abiotic stress response by affecting anthocyanin synthesis. Expression of several anthocyanin synthesis genes, such as *ANS*, *DFR*, *F3′H*, *UGT75C1* and *UGT78D2* increase significantly in *Pro35S:MIR156* plants. Among these genes, *DFR* is most sensitive to miR156-SPL activity. Besides SPLs activity, the MYB-bHLH-WD40 complex is also required for activating the expression of DFR. A yeast two-hybrid assay illustrated that SPL9 competently binds to PAP1 with TTT8 to prevent the formation of MYB-bHLH-WD40, leading to repression of *DFR* expression. Based on those results, when plants encounter abiotic repress (at least salt and drought), miR156 is induced and repress SPLs gene expression. Then, PAP1 interacts with TTT8 and TTG1 to form MYB-bHLH-WD40 complex, which binds to the promoter region of anthocyanin biosynthesis genes (ABGs), activating anthocyanin biosynthesis (Figure 3a). When favorable conditions recover, miR156 reduce while SPL9 increase and then bind to PAP1, interfering with MYB-bHLH-WD40 complex formation to repress ABGs expression [40]. This miR156-SPL-mediated abiotic stress response mechanism is functionally conserved in *Oryza sativa*. However, further knowledge of the relationship between other abiotic stressors are still required, including climate change, UV damage, nutrient deficiency and miR156-SPL-mediated vegetative phase transitions.

#### 3.2.2. miR156 Participate in Defense against Invading Pathogens

In contrast to abiotic stress response, older plants display more resistance to pathogens. Herbivores insects favor younger plants over older or mature ones, which is referred as the plant vigour hypothesis [41]. Recently, Mao et al., [39] proposed a compensating model whereby defense compounds accumulation against JA-response attenuation during plant maturation. Their model based on the contradiction that mature plants have more insect and pathogen resistance, whereas the accumulation of JA response reduced with plant age. Overexpression of SPL9 retards insect resistance, demonstrating miR156-SPL9 negatively affects insect resistance. Further yeast two-hybrid experiments demonstrated that SPL9 interact with numbers of JAZ proteins, which act as a JA signal repressor. This JAZ protein has two conserved domains, the ZIM-domain (N-terminal) and Jas domain (C-terminal). The Jas domain interacts with other transcriptional factors, including COI1, which mediates degradation of JAZs by the 26S proteasome. Interestingly, the ZIM-domain of JAZ contains a sensor response to an age cue. Increasing abundance of SPL9 induces JAZ accumulation and weakens the interaction between COI1 and JAZ. This demonstrates that SPL9 promotes JAZ accumulation by binding to the ZIM-domain, maintaining its stability by preventing COI1-mediated degradation, resulting in repression of JA’s response as the plant ages (Figure 3b). However, old plants display higher resistance against insects and pathogens, whereas they have lower JA response ability. Also, Mao et al. [39] observed the accumulation of Glucosinolate (GLSs) as plants age to strengthen insect resistance. However, the accumulation of GLSs might not be governed by the miR156-SPL module. Therefore, they hypothesize that in the juvenile phase, plants are too young to accumulate enough defense factors, thus a high JA response is essential for protecting the plant against insects and pathogens. As plants age, constitutive defense factors accumulate to the threshold by which is enough to protect plants, then JAZ proteins accumulate to repress JA signals. Although JA acts as a major defense hormone against insects and pathogens, JA inhibited the development through with interfering other phytohormones such as auxin and GA signals [42,43,44]. Thus the balance between plant age and JA response might be the strategy to ensure successful development in *Arabidopsis*.

## 4. A Big Portrayal of miR156 Regulation by Deep Sequencing

Regarding the features of miRNAs and their tremendous number in plants, high-throughput sequencing (HTS) platforms enable whole genome or transcriptome searches for miRNA with unprecedented coverage and depth compared with traditional Sanger sequencing. Over the past decade, genome sequencing, transcriptome sequencing (RNA sequencing), and sRNA sequencing (sRNA-seq) have been widely used to elucidate the mechanisms underlying diverse biological processes. The HTS platform can help to address some unique questions, such as whether the miR156-SPL module plays roles in other fundamental processes through interaction with other miRNAs or independently in *Arabidopsis*.

### 4.1. The miR156-SPL Module Linking Multiple Pathways

In recent years, many miRNAomes of *Arabidopsis* have been identified and characterized through HTS to clarify that miR156 and its target play roles in multiple fundamental processes, such as in the development of embryos and siliques, abiotic and biotic stress response and nutrient deficiency responses (Table 1). Inducing or elevating the amounts of miR156 gene expression are detected in such processes as abiotic stress, elevated temperature, nutrient deficiency and some parts of embryo and silique development [9,10,12,13,15,17,18,45,46,47]. In contrast, the downregulation of miR156 genes is identified in viral defense, the elevation of CO_2_ content and embryo pattern formation [11,12,48]. Interestingly, the expression pattern of miR156 genes when plants encounter abiotic stress consists of the results from molecular experiments, demonstrating their reliability of HTS. SPLs are also identified as the main target of miR156 by HTS, as is expected. But several novel targets, such as GAI, EMB140, and PMEI, are detected as putative targets of miR156 in phosphate deficiency response and embryogenesis, respectively (Table 1). However, the reliability as miR156 targets and their actual roles of these novel targets still require evidence from molecular experiments.

### 4.2. Inference of miR156-miRNAs Regulate Network in Arabidopsis

One of the most valuable advantages of HTS technology is that one can identify abundant different expressions of miRNAs at one time in comparison to the traditional manner. Numbers of conserved miRNAs are identified associated with miR156 in those processes described above. Among these, correlated miRNAs—some of them including miR160, miR164, miR172, miR157, miR167, miR399, miR778, and miR827—are participating in more than three processes. Based on this observation, we painted a diagram for illustrating the miR156-miRNAs-associated network in multiple biological processes of *Arabidopsis*. miR160, which targets transcripts of auxin-responsive factors (ARFs) correlated with miR156 in almost six processes (Figure 4). miR160 has dual expression pattern comparison with that of miR156 in different processes. For example, heat stress induces expression of both miR156 and miR160, whereas TCV infection induces miR160 while repressing miR156 expression. Another auxin signals responsive miRNA, and miR164 also takes part in both stress response and development process. This connection between miR156 and miR160/miR164 demonstrating a potential balance between plant age and auxin signals transduction. In contrast, miR172 mainly participated in the flowering-related process. Both the elevation of CO_2_ and temperature have effects on flowering time, thus it is reasonable to detect the alteration of miR172 expression. However, the contrary of miR172 expression in response to elevated CO_2_ (up) and temperature (down) requires more investigation, since both elevations induce early flowering. Obviously, these miR156-miRNAs regulation networks need more information and refinement with the help of high-throughput sequencing and molecular experiments in the future.

### 4.3. Limitation of HTS for Idenftied Function of miR156

The past two decades have witnessed the increasing amount of studies utilizing next-generation sequencing to discover and characterize microRNAs in different processes [50]. Among those, different expressions of several miR156 genes, pri-miR156 and mature miR156 have been detected in several processes. However, an inevitable limitation of miRNAs sequencing refers to the fact that the reads number of the miRNA gene does not necessarily correlate with the actual abundance of mature miRNAs. For instance, two miR156 genes, *MIR156E* and *MIR156G*, are induced to a significant level under N-starvation conditions, whereas the mature miR156 abundance reduced comparing to nutrient-replete conditions [15]. Another limitation refers to the fact that a different expression pattern of miR156 could be detected in the same experimental conditions. As we described above, mature miR156 does not increase under N-starvation conditions [15], while the induction of mature miR156 is reported by the other two works [17,45]. These discrepancies might be the result from the biases caused by sample preparation and different sequencing strategies [50]. Although some limitations must be addressed, high-throughput sequencing provides novel insight and broadens our vision for the role of miR156-SPLs in *Arabidopsis*.

## 5. Conclusions

Flowering plants undergo several continuous development processes after germination. The coordinated transition of plants from juvenile to adult phases is critical for flowering. This transition is mainly mediated by the miR156-SPL module. With aging, the abundance of miR156 decreases continuously, whereas its target genes, SBP/SPLs, increase. These changes have been regulated through different pathways, including the GCT/CCT pathway, the sugar-HXK1 pathway and require epigenetic modification. The miR156-SPL-mediated plant age pathway is involved in many other biological processes by impacting downstream targets at transcript level or through protein interaction. These downstream processes include plant development from embryo to silique, response to abiotic and biotic stress, nutrient deficiency and climate change. Based on the investigation from both molecular and high-throughput sequencing experiments, we draw a diagram for a miR156-SPL module regulation network in *Arabidopsis*. Eight conserved miRNAs cooperated with miR156 in at least three different processes. Some miRNAs, including miR160 and miR164, target auxin signals involved in both development and stress response, while miR172 mainly participates in the flowering-related processes to ensure the proper flowering time of *Arabidopsis*. Although further experimental verifications for the connections between miR156-SPL and other miRNAs are needed to illustrate how such phase change-related genetic networks play a crucial role in modulating other fundamental processes, utilizing of high-throughput sequencing will help us to further understand the miRNA regulation network in *Arabidopsis*.

## Figures and Tables

**Figure 1 ijms-20-06166-f001:**
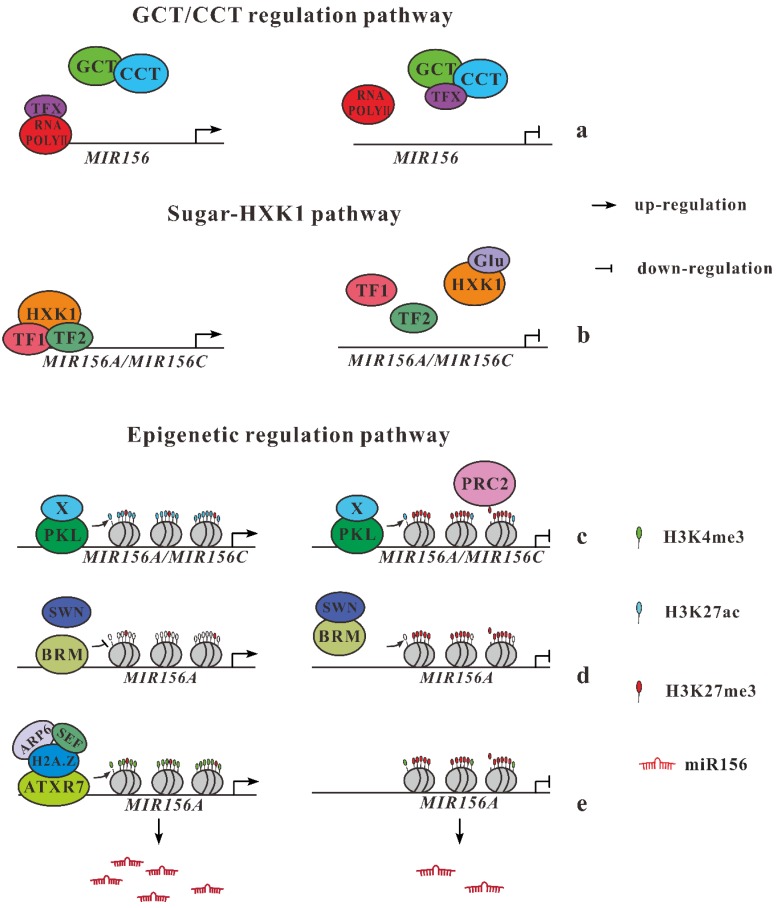
Regulation pathway of *MIR156* gene expression in *Arabidopsis*. (**a**) GCT/CCT regulate expression of *MIR156* genes through competitive interactive with unknown transcript factor, inactivating RNA polymerase II. (**b**) Under low sugar conditions (left), HXK1 transcript complex bind into promoter of *MIR156A/MIR156C*. Then glucose incorporate with HXK1 (right), inhibiting MIR156 gene expression. (**c**) The decreases in the expression of MIR156A/MIR156C are regulated by the levels of histone H3 lysine 27 methylation (H3K27me3). PKL reduces the amount of H3K27ac continuously with a histone deacetylase (X). PRC2 binds to the promoter of *MIR156* gene then methylates H3K27 when the levels of H3K27ac no longer inhibiting PRC2 binding. (**d**) BRM activates *MIR156A/MIR156C* expression by decreasing levels of H3K27me. Then SWN bind to BRM, releasing BRM from the promoter of *MIR156A/MIR156C* to increase H3K27me abundance leading to repression of miR156 expression. (**e**) At the initial juvenile phase, ARP6 interacts with SEF, exchanging H2A to H2A.Z, which is the deposition of H3K4me3 at the promoter region of *MIR156* locus with the help of ATXR7.

**Figure 2 ijms-20-06166-f002:**
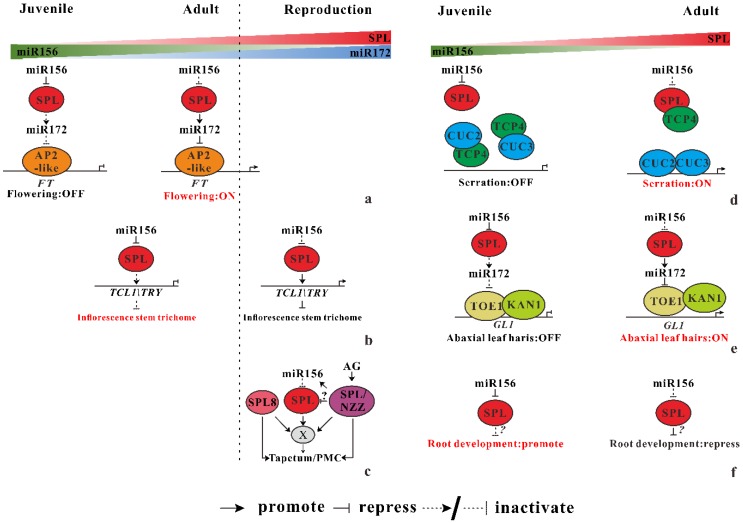
miR156 regulates multiple development traits in *Arabidopsis*. (**a**) When plants age, the amount of miR156 decreases, while its targets’ SPL increases, directly promoting miR172 expression. Then, the AP2-like transcript factor no longer inhibits the *FT* gene and plant flowering. (**b**) miR156-SPL module active trichome distribution at inflorescence stem by direct bind to *TCL/TRY* gene. (**c**) After flowering, miR156-targeted SPLs and non-targeted SPL8 secure male fertility by interacting with SPL/NZZ or independently regulating cell division and differentiation to proper cell type (Tapetum/PMC). X indicates an unknown regulator acts as a direct downstream target. (**d**) With plant age, miR156-SPL-miR172 targets TOE1 to break its interaction with KAN1, leading to abaxial leaf hair initiation. (**e**) In the juvenile phase, TCP4, which is the target of miR319, suppresses the dimerization of CUC3 and CUC2, which is the target of miR164. Then, SPL competes with CUCs for SPL interaction, thus allows CUC2–CUC3 dimerization and increase leaf serration and complexity. (**f**) miR156-SPLs repress lateral root development with plant age. The direct downstream target of SPLs remains unknown.

**Figure 3 ijms-20-06166-f003:**
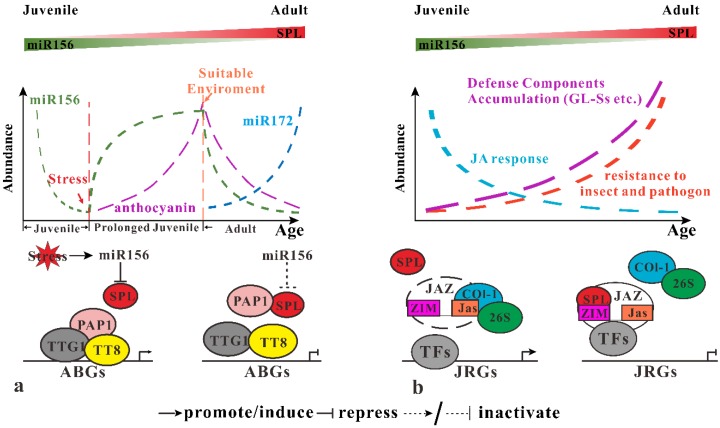
miR156-SPLs modulate response to abiotic/biotic stress. (**a**) miR156-SPL module regulates the response to abiotic stress through their effect on anthocyanin biosynthesis. When plants encounter exogenous stress, miR156 is induced to repress SPL, preventing its interaction with PAP1 so that PAP1 bind to TTT8 and TTG1 to form MYB-bHLH-WD40 complex, which activates anthocyanin biosynthesis genes (ABGs) expression. When environment conditions recover, miR156 abundance decreases to activate SPLs, which break the MYB-bHLH-WD40 complex by competent binding to PAP1 with TTT8. (**b**) miR156-SPLs regulate infect and pathogen defense by regulation JA response. SPLs bind to the ZIM-domain of JAZ protein to prevent it from COI1 mediated 26s protease degradation to repress expression of JA-responsive genes (JRGs). Meanwhile, several defense components accumulate with plant age to increase insect and pathogen defense for old plants.

**Figure 4 ijms-20-06166-f004:**
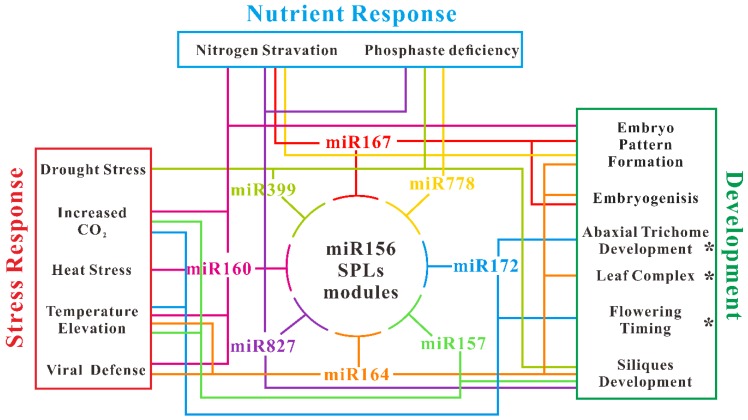
Diagram of miR156-miRNAs regulation network. The illustration of this network is based on both molecular experiments and HTS data described in the text. The asterisk indicates a verified connection between miR156 and other miRNAs.

**Table 1 ijms-20-06166-t001:** Multiple roles of miR156 identified by high-throughput sequencing in different biological processes.

Biological Process	Conserved miRNAs	Novel miRNAs	Expression Pattern	Target Genes	Correlated miRNAs	References
Drought stress	123	NA	Up	SPL9	2	[9]
Heat stress	6 (5 families)	NA	Up	NA	4	[10]
Viral defense	30 (17 families)	29	Down	SPLs *	15	[11]
Elevated CO_2_ content	18 (8 families)	1	Down	SPL10	4	[12]
Elevated temperature	36 (14 families)	4	Up	SPL2 SPL3 SPL11 SPL13 SPL15	6	[12]
Phosphaste deficiency	55 (23 families)	NA	Up (root)	SPLs * GAI	9	[17]
Nitrogen starvation	41 (34 families)	9	Up	SPLs * GAI	10	[45]
	20 (8 families)	NA	Up	NA	6	[15]
Iron homeostasis	NA	NA	Up	SPL9 SPL15	NA	[13]
Embryogenisis	59 (47 families)	NA	Down	SPL2 SPL10 SPL15 EMB140 PMEI	4	[48]
Embryo pattern formation	39 (13 families)	NA	Up	SPL2 SPL3 SPL10 SPL11	12	[18]
	15 (10 families)	NA	Up	SPL10 SPL11	9	[46]
Siliques development	15 (9 families)	5	Up	SPL15	8	[47,49]

The number in the parentheses indicate the number of miRNAs gene families. * SPLs include SPL2, SPL3, SPL4, SPL9, SPL10 and SPL13B. NA: not mentioned in the reference.

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
