# Peer review of "A Regulatory Network for miR156-SPL Module in Arabidopsis thaliana"

_ijms, 2019, doi:10.3390/ijms20246166_

Round 1

Reviewer 1 Report

In general, this review looks better than the first version. However, I would suggest a grammar revision done by professionals. That would improve the paper's value, and would help the readability. I have some minor changes to be addressed:

Minor changes

Line 59: please rephrase "Many studies have begun to study".

Lines 172-174: Please rephrase and correct the sentence. 

Lines 203 and 246: Please, make sure you introduce an extra line before the titles to help readers distinguish between sections. 

Please, reformat Table 1 to a more reader friendly version. If possible, make it fit in one page. 

Author contributions: What does it mean "RW planned and designed the research."?

Author Response

Response to Reviewer 1 Comments

Point 1: Line 59: please rephrase "Many studies have begun to study".

Response 1: we have rephrased this sentence as “How age regulates the temporal decrease of miR156 expression has been documented in different studies” in lines 66-67 in the revised manuscript.

Point 2: Lines 172-174: Please rephrase and correct the sentence. 

Response 2: We rewrite this sentence to “As demonstrated by chromatin immunoprecipitation results, miR172 is directly activated by SPL9, and redundancy by SPL10 protein, promoting juvenile-adult phase transition in spl transgenic plants” in line 191-193 of the revised version.

Point 3: Lines 203 and 246: Please, make sure you introduce an extra line before the titles to help readers distinguish between sections. 

Response 3: We check over the whole manuscript and make sure we introduce an extra line before each title in the revised manuscript.

Point 4: Please, reformat Table 1 to a more reader friendly version. If possible, make it fit in one page. 

Response 4: We reformatted Table 1 and we list the number of correlated miRNAs instead of all of their names in the revised version. Then those miRNAs that participated in more than three different biological processes were chosen for painting a miR156-miRNAs network illustrated as Figure 4. We hope that the new reformatted table 1 will be more friendly for readers.

Point 5: Author contributions: What does it mean "RW planned and designed the research."?

Response 5: We corrected the author’s contributions: “Conceptualization, RW; writing-original draft preparation, CZ, MY, MS, RW; writing-reviewing and editing, CZ and RW; supervision, RW.” in the revised documents.

Reviewer 2 Report

The manuscript presented by Zheng et al is interesting and valuable review related to regulatory module of miRNA156-SPL. Generally, this Review gives an comprehensive overview on functionally role of miRNA156 in plant developmental transition and other basic biological processes in plants life cycle.

I have just minor comments for Authors listed below:

Introduction is the 1st number in overall numbering the section (not a 0. as in the manuscript, page 1 line 27). I am not sure is this part necessary at all in the Review since the concept of all work is presented in the Abstract. I would skip Introduction unless is it required by journal. There is a lot of typos therefore I ask Authors for carefully revising the text and check it in detail (e.g. no space between word and the brackets page 1 line 29, 30, and in many places in the manuscript; glutose instead of glucose; histon instead of histone in the Figure 1 description; regulate instead of regulates page 8 line 268; 289) The overall editing of whole manuscript structure is needed (there is no interline between the text and the title of subsection which makes hard to distinguish between text fragments (e.g. page 7 line 203; 246) Figure 3 should be enhanced in terms of quality and resolution Table 1 needs attention, it is not in publishable format in the present form in my opinion Figure 4 - the classification of stresses into categories 'stress response' and 'climate change' is little bit controversial to me. Since heat stress together with drought stress are the consequences of climate changes.... Conclusions section should be shortened and I propose bullet pointed way of presenting this section with the use of the most important statements - take home messages.

Author Response

Response to Reviewer 2 Comments

Point 1: Introduction is the 1st number in overall numbering the section (not a 0. as in the manuscript, page 1 line 27). I am not sure is this part necessary at all in the Review since the concept of all work is presented in the Abstract. I would skip Introduction unless is it required by journal.

Response 1: We refer to some published IJMS papers and check the introduction for authors. The Introduction section is required in this journal. Then we correct it as 1st number in the revised manuscript.

Points 2: There is a lot of typos therefore I ask Authors for carefully revising the text and check it in detail (e.g. no space between word and the brackets page 1 line 29, 30, and in many places in the manuscript; glutose instead of glucose; histon instead of histone in the Figure 1 description; regulate instead of regulates page 8 line 268; 289). The overall editing of whole manuscript structure is needed (there is no interline between the text and the title of subsection which makes hard to distinguish between text fragments (e.g. page 7 line 203; 246)

Response 2: We are sorry for those typos and we check the manuscript word by word carefully and we also check the spaces between words and brackets. Those typos are corrected and we make sure that there are spaces between words and brackets and that there are interline between the text and titles of subsections in the revised version.

Point 3: Figure 3 should be enhanced in terms of quality and resolution

Response 3: We improved the quality and enhanced the resolution to 600 dpi for not only figure 3 but also all figures to improve their readability.

Point 4: Table 1 needs attention, it is not in publishable format in the present form in my opinion

Response 4: We reformatted Table 1 and we list the number of correlated miRNAs instead of all of their names in the revised version. Then those miRNAs that participated in more than three different biological processes were chosen for painting a miR156-miRNAs network illustrated as Figure 4. We hope that the new reformatted table 1 will be more friendly for readers.

Point 5: Figure 4 - the classification of stresses into categories 'stress response' and 'climate change' is little bit controversial to me. Since heat stress together with drought stress are the consequences of climate changes

Response 5: We delete the categories ‘climate change’ and we clustered both elevated CO2 content and elevated temperature into ‘stress response’ categories. Although heat together with drought stress results from elevated temperature, the underline mechanisms for miR156-miRNAs network response to them might be different. Therefore, we kept them all in the ‘stress response’ categories in the revised figure 4.

Point 6: Conclusions section should be shortened and I propose bullet pointed way of presenting this section with the use of the most important statements - take home messages.

Response 6: We shorted the conclusion section and only the most important information was maintained in the revised version. We hope they are more friendly to readers.

This manuscript is a resubmission of an earlier submission. The following is a list of the peer review reports and author responses from that submission.

Round 1

Reviewer 1 Report

In the paper “A miRNA-regulatory network for vegetative phase 2 change by deep sequencing”, the authors describe how miRNAs regulate the transition between juvenile and reproductive phases. They mainly focus on miR156 and Arabidopsis, but include data from other species.

Major comments

My main concern is that this review paper ended up being an incomplete review of miR156 and its functions, but I do not see the interest for the field to have another review paper of miR156. Also, the authors present “new” results in the second part of this article, where they use available data to generate co-expression networks and functional diagrams. I have the feeling that these two sections belong to two “incomplete” different papers that were just pasted together.

In addition, I am unable to see figure 1 properly. All I see is a black background with few colored bubbles…

Missing references

If this is considered a review of mir156 here are some missing key references, among others:

- miR156-
SPLmodules regulate induction of somatic embryogenesis in citrus callus
- The miR156/SPL Module, a Regulatory Hub and Versatile Toolbox, Gears up Crops for Enhanced Agronomic Traits
- miR156/SPL10 Modulates Lateral Root Development, Branching and Leaf Morphology in Arabidopsis by Silencing AGAMOUS-LIKE 79

Reviewer 2 Report

Zheng et al. in the manuscript entitled ‘A miRNA-regulatory network for vegetative phase change by deep sequencing’ reviewed recent data regarding the role of miRNA156-SPL module in regulation the transition from juvenile to adult stage. Moreover, Authors perform computational analyses of publicly available NGS data and demonstrated network where miRNA156 has a central position.

From my point of view, it makes hard to distinguish between classical Review article (and Authors designated their article as such) and Research article…. Presented manuscript gives more than you can find in the literature and thus can be recognized as Research (even Authors in the ‘Authors contribution’ section mentioned who planned and performed research), on the other side it lacks crucial section for Research paper – Material and Methods…. This is my major comment to the manuscript. Authors should decide and prepare manuscript in more proper way.

Minor comments:

Figure 1 needs attention since it is totally black and thus not usable In the line 85 page 4 – Yu et al and Yang et al references need a year in brackets Figure 2 – maybe including letters (A, B, C…) for each panel will make the figure more clear In the line 124 page 6 – typo in ‘promoting’ word Line 145-152 page 6 this paragraph seems to be unnecessary since NGS techniques are broadly known and serves as basic molecular method nowadays Table 1 – why Arabidopsis thaliana is bolded and not italic whereas other species are? Title of Table 1 suggests that we find there ‘examples of miRNA’ but there is no such information. The title is misleading and the table without examples of identified miRNAs seems to be not informative Line 197 page 9 – ‘miR156-SPL molecule’ is not appropriate term, probably it is a typo, should be module or complex but not the molecule Line 199-202 – authors mentioned construction of co-expression networks but there is no methodology, no information about statistics and software/package used for construction. Also, this part is more proper for Research Article than Review as I mentioned in the first paragraph of this review. Line 214 page 9 – what Authors understand as ‘ancient’ miRNA? Is it highly conserved? Line 229 – the phrase ‘we found’ is another premise to consider the research paper based on obtained data than Review Line 245 – if something ‘sheds a light on’ something it is a result, and the results should be published as Research Article…

I would like to highly recommend reconsideration of the type of the Article. Maybe including the network with explanation of its construction would be good idea for Review Article, but I’d rather encourage to enrich the manuscript with other data and publish it as Research.